**Fungal Genetics and Genomics**

# Gene loss and cis-regulatory novelty shaped core histone gene evolution in the apiculate yeast *Hanseniaspora uvarum*

Max A.B. Haase (ID) ,[1,3,]* Jacob L. Steenwyk (ID) ,[2] Jef D. Boeke (ID) [1]

[1]Institute for Systems Genetics and Department of Biochemistry and Molecular Pharmacology, NYU Langone Health, 435 E 30th St, New York, NY 10016, USA
[2]Howards Hughes Medical Institute and the Department of Molecular and Cell Biology, University of California, Berkeley, Berkeley, CA 94720, USA
[3]Present address: Department of Mechanistic Cell Biology, Max Planck Institute of Molecular Physiology, Dortmund, Germany

*Corresponding author: New York University, Institute for Systems Genetics, 435 East 30th Street, Room 923B, New York, NY 10016, USA. Email: max.haase@mpi-dortmund.mpg.edu

Core histone genes display a remarkable diversity of cis-regulatory mechanisms despite their protein sequence conservation. However, the dynamics and significance of this regulatory turnover are not well understood. Here, we describe the evolutionary history of core histone gene regulation across 400 million years in budding yeasts. We find that canonical mode of core histone regulation—mediated by the trans-regulator Spt10—is ancient, likely emerging between 320 and 380 million years ago and is fixed in the majority of extant species. Unexpectedly, we uncovered the emergence of a novel core histone regulatory mode in the *Hanseniaspora* genus, from its fast-evolving lineage, which coincided with the loss of 1 copy of its paralogous core histone genes. We show that the ancestral Spt10 histone regulatory mode was replaced, via cis-regulatory changes in the histone control regions, by a derived Mcm1 histone regulatory mode and that this rewiring event occurred with no changes to the trans-regulator, Mcm1, itself. Finally, we studied the growth dynamics of the cell cycle and histone synthesis in genetically modified *Hanseniaspora uvarum*. We find that *H. uvarum* divides rapidly, with most cells completing a cell cycle within 60 minutes. Interestingly, we observed that the regulatory coupling between histone and DNA synthesis was lost in *H. uvarum*. Our results demonstrate that core histone gene regulation was fixed anciently in budding yeasts, however it has greatly diverged in the *Hanseniaspora* fast-evolving lineage.

Keywords: gene regulation; evolution; *Hanseniaspora*; yeast; histones; gene loss; genomics; chromatin; chromosome

## Introduction

Chromatin structure and function are critical for essential processes, including DNA replication, chromosome division, DNA damage repair, and transcription (Wei *et al.* 1999; MacAlpine and Almouzni 2013; Venkatesh and Workman 2015; Hauer *et al.* 2017; Haase, Lazar-Stefanita, *et al.* 2023, Haase, Ólafsson, *et al.* 2023; Lazar-Stefanita *et al.* 2023). The basic unit, the core nucleosome particle, is an octameric protein complex made up of an H3–H4 tetramer flanked by 2 separate H2A–H2B dimers that together wrap ~146 bp of DNA (Luger *et al.* 1997). Core histone genes are often present at multiple copies encoded as gene clusters in eukaryotic species. In the yeast *Saccharomyces cerevisiae*, each core histone is encoded by 2 paralogous genes that are arranged in the genome as divergently transcribed clusters (Fig. 1a), which either encode H2A–H2B (*HTA1B1* and *HTA2B2* loci) or H3–H4 (*HHF1T1* and *HHF2T2* loci; Eriksson *et al.* 2012). Reflecting their fundamental importance, the nucleosome structure and the primary amino acid sequence of histones are highly conserved across eukaryotes (Malik and Henikoff 2003); the yeast paralogous histone proteins are identical (H3/4 and H2B) or near identical (H2A). Tight control of the regulation of core histones throughout the cell cycle ensures the proper function of DNA-templated processes

(Eriksson *et al.* 2012). Proper stoichiometry between nucleosomes and total DNA content is partly controlled through replication-coupled synthesis of histones (Robbins and Borun 1967), evidenced by experimental inhibition of DNA synthesis leading to rapid repression of histone synthesis (Osley 1991; Rattray and Müller 2012; Bhagwat *et al.* 2021) Moreover, misexpression of histones outside of S-phase characteristically leads to cellular toxicity and growth arrest (Kurat, Recht, *et al.* 2014).

Despite the evolutionarily conserved nature of cycle-dependent expression and function of histones (Jensen *et al.* 2006), the cis-regulatory mechanisms used to achieve precise control of core histone expression are diverse across eukaryotes (Mariño-Ramírez *et al.* 2006). In the budding yeast *S. cerevisiae*, specific S-phase expression is driven by both positive and negative regulatory mechanisms (Eriksson *et al.* 2012; Kurat, Recht, *et al.* 2014; Fig. 1b). The primary transcriptional regulation is found within the histone control regions (defined as the intervening sequence between the 2 divergently transcribed histone genes) and is primarily mediated by the transcription factor Spt10 (Fig. 1a), a putative acetyltransferase that positively induces transcription through the histone Upstream Activating Sequence (UAS; Dollard *et al.* 1994; Eriksson *et al.* 2005). Prior work has shown that this Spt10-mode is deeply conserved across yeast phylogeny

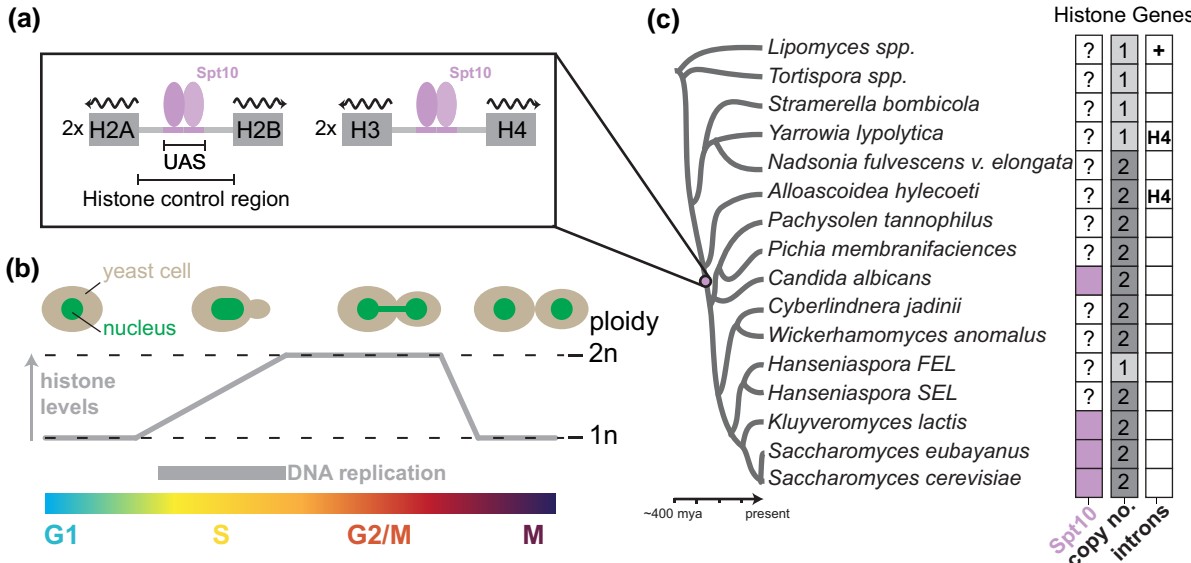

**Fig. 1.** Histone gene regulation in *Saccharomyces cerevisiae* and related species. a) The cell cycle expression of histone in *S. cerevisiae* is positively regulated by the trans-regulator Spt10, which binds to the upstream activating sequences within the histone control regions. Each of the histone gene clusters (H2A–H2B and H3–H4) is present at 2 copies per genome. b) Histones are primarily expressed during S-phase of the cell cycle (see these reviews for details; Eriksson *et al.* 2012; Kurat, Lambert, *et al.* 2014). c) Spt10-mode of histone regulation is conserved and was at least present in the common ancestor of *S. cerevisiae* and *C. albicans* (Mariño-Ramírez *et al.* 2006).

(Mariño-Ramírez *et al.* 2006); however, the specific origins and emergence of this regulatory mode still need clarification (Fig. 1c). In addition, negative regulatory feedback is achieved through various mechanisms (Eriksson *et al.* 2012), including various histone chaperones (HIR complex, Atf1, and Rtt106), RSC chromatin remodelers, and by less well-conserved DNA–protein interactions (i.e. NEG region; Eriksson *et al.* 2012). Despite these advances, the specific origins and evolutionary dynamics of histone regulatory modes in budding yeasts remain largely unknown.

Mechanisms of core histone regulation are particularly enigmatic in the genus *Hanseniaspora*, a group of bipolar budding yeasts belonging to the order Saccharomycodales (Groenewald *et al.* 2023). The evolutionary history of *Hanseniaspora* spp. is marked by a burst of rapid evolution and the loss of numerous conserved genes, including those associated with cell cycle processes and genes involved in DNA repair (Steenwyk *et al.* 2019). As a result, some *Hanseniaspora* spp. have genome sizes of around 8–9 megabases and encode approximately 4,000 genes (Steenwyk *et al.* 2019). In contrast, *S. cerevisiae* has a genome size of roughly 12 megabases and encodes approximately 6,000 genes (Goffeau *et al.* 1996). The degree of rapid evolution and gene loss is more pronounced in 1 *Hanseniaspora* lineage compared to the other and are thus termed the faster-evolving lineage (FEL) and slower-evolving lineage (SEL), respectively. However, the dearth of established tools for genetic manipulation of *Hanseniaspora* spp. has stymied our understanding of its cell biology and genetics (Schwarz *et al.* 2022; Heinisch *et al.* 2023), including how gene loss has (re)shaped cell cycle processes. In contrast, much more is known about *Hanseniaspora* ecology. *Hanseniaspora* spp. are abundantly present on various fruits (e.g. grapes) and associate with various insects (such as *Drosophila* spp.), which are attracted to volatile aromatic compounds produced by *Hanseniaspora* spp. (Hamby *et al.* 2012; Becher *et al.* 2018; Saubin *et al.* 2020). *Hanseniaspora* has also gained interest in biotechnology applications, such as expanding the sensorial complexities of fermented products (Steensels and Verstrepen 2014) and as natural

biocontrol agents (Rueda-Mejia *et al.* 2023). Among *Hanseniaspora* spp., *Hanseniaspora uvarum*, a species in the FEL, has become a focus for research and biotechnological development (Badura *et al.* 2021, 2023; Heinisch *et al.* 2023; Van Wyk *et al.* 2023).

Here, we explore the evolution of core histone genes and their cis-regulatory evolution in a panel of yeast that span the diversity of the Saccharomycotina subphylum and conduct an in-depth computational and molecular investigation of *Hanseniaspora*. Using a histone replacement assay, we find evidence suggesting that *H. uvarum*'s H2A–H2B dimer and, surprisingly, its histone control regions are incompatible in *S. cerevisiae*. Examination of the cis-regulatory changes underlying the histone control region incompatibility revealed that the ancestral Spt10-mode was replaced with a derived Mcm1-mode of regulation in *H. uvarum* and other FEL species. Moreover, we show that the function and regulatory network of *H. uvarum*'s Mcm1 is conserved with *S. cerevisiae*, suggesting that the histones were rewired into a novel regulatory paradigm in *H. uvarum*. Characterizing cell cycle dynamics and the timing of histone synthesis in single cells of *H. uvarum*, we uncovered a rapid division with a doubling time of ∼60 minutes, and surprisingly, we found that histone and DNA synthesis status were decoupled, unlike the case in *S. cerevisiae*. These findings uncovered unexpected novelty in a hitherto conserved and fundamental cellular process. More broadly, this work lays the foundation for future genetic investigations into the highly divergent genus of *Hanseniaspora*.

## Materials and methods
### Yeast strains and plasmids
All strains and plasmids are listed in Supplementary Tables 1 and 2, respectively. Both *S. cerevisiae* and *H. uvarum* strains were grown in standard yeast media (YPD or SC) at 30°C. *S. cerevisiae* strains were transformed using standard lithium acetate procedures (Gietz and Schiestl 2007). We transformed *H. uvarum* by electroporation following a previously published method with

modifications (Heinisch *et al.* 2023). Briefly, a 50-mL overnight culture ($A_{600} \approx 0.1$) was grown at 30°C for 16 hours at 210 rpm. The next day, the density of the culture was checked; if $A_{600} \approx$ 3.5–4.0, we immediately proceeded to the following steps; however, if $A_{600} > 4.0$, we diluted the culture and let it grow for an appropriate amount of time. Cells were then collected, washed 1× in sterile water, resuspended in lithium acetate buffer (10-mM Tris, pH 8.0, 1-mM EDTA, 100-mM lithium acetate, 10-mM DTT), and incubated for 1 hour at room temperature with agitation. Next cells were collected, washed in ice-cold sterile water, and then washed 2× in ice-cold 1-M sorbitol. Finally, cells were collected and resuspended in 500 µL of ice-cold 1-M sorbitol (this mixture can be frozen at −80°C for later transformation). For 1 transformation, we used an aliquot of 100 µL of the cell suspension. A maximum of 10 µL of DNA (~1 µg) was placed into a 2-mm cuvette, and cells were added onto and mixed by lightly tapping the cuvette; each cuvette was then incubated at room temperature for 10 minutes. Next, samples were electroporated using a Bio-Rad MicroPulser with the "SC2" default settings. After electroporation, the cell suspension was diluted with 1-mL fresh YPD medium, transferred to a 1.5-mL centrifuge tube, and incubated with rotation for 3–4 hours. Lastly, cells were collected, and the transformation was plated to YPD + 400 µg of hygromycin B.

## Histone gene presence and absence in *Hanseniaspora*

We initially searched a set of published *Hanseniaspora* genomes and 4 outgroup species (*S. cerevisiae*, *Kluyveromyces lactis*, *Cyberlindnera jadinii*, *Wickerhamomyces anomalus*) for histone genes with BLASTP, using the protein sequences of *S. cerevisiae*'s histones as the query. Each hit was manually inspected and verified to remove erroneous hits such as histone variants H2A.Z or CenH3 (Cse4). For species of the FEL, we determined which of the histone clusters were lost through synteny analysis, comparing *Hanseniaspora* genomes to *S. cerevisiae* and the inferred pre-WGD ancestor using the Yeast Gene Order Browser (Byrne and Wolfe 2005).

## Dual-plasmid histone shuffling and plasmid cloning

We modified our plasmid shuffling tool set for histone humanization in order to shuffle in the *Hanseniaspora* histone genes and their control regions. Briefly, a histone shuffle strain (yMAH302), which has all 8 chromosomally encoded core histone genes deleted and a single copy of each core histone encoded on a counter-selectable plasmid, is transformed with an incoming set of histone genes. By counterselection of the *URA3* marker with 5-FOA, the cells are forced to use the incoming set of histone genes (*TRP1* marker), and the assay readout is cell growth. We used the low-background "Superloser" shuffle plasmid to reduce the frequency of spontaneous *ura3* mutants that lead to erroneous 5-FOA$^R$ colonies (Haase *et al.* 2019). For our histone shuffling assays, we used the native histone DNA sequences from *Hanseniaspora* species, as we observed that codon-optimized versions complemented worse (data not shown). We followed a general workflow for plasmid shuffle assay as previously published (Haase *et al.* 2019, Haase, Ólafsson, *et al.* 2023).

## Core histone control regions analysis

Core histone control regions were defined as the intervening sequence between the divergently transcribed histone genes. Regulatory motif enrichment analysis was done as previously described with the following variations (Haase *et al.* 2021). Using AME (Analysis of Motif Enrichment, default options), we searched each histone control region for the canonical Spt10 UAS from *S. cerevisiae*. Next, using MEME (maximum width = 24, site distribution = anr), we identified the 2 motifs, "Spt10-like" and "Mcm1-like", from a search of all histone control region sequences—using all *Hanseniaspora* species and the outgroup species as shown in Fig. 3c. We then again used the AME search (now using either the "Spt10-like" and Mcm1-like" motif) to determine which species each motif was associated with, observing the strong enrichment of the "Mcm1-like" motif in the *Hanseniaspora* FEL.

## Histone control region UAS replacement assays

We precisely deleted, or replaced with a consensus Spt10 UAS, the putative Mcm1 binding sites from the *Huva*HCR (both H2A–H2B and H3–H4). These constructs were then used for plasmid shuffle assays to assess their function. For measurements of GFP expression, we used a ubiquitin-N-degron GFP reporter (Houser *et al.* 2012) integrated at the HO locus with either no upstream control region or the indicated histone control region, as shown in Fig. 3d. Total cellular fluorescence was determined from single cells from a series of images acquired on an EVOS M7000 imaging system.

## Mcm1 motif discovery of Mcm1-regulated orthologs in *H. uvarum*

We identified a list of Mcm1-regulated genes in *S. cerevisiae* (Spellman *et al.* 1998) and extracted the protein sequences of these genes. We then used BLASTP to search the genome of *H. uvarum* for orthologs of these Mcm1-regulated genes. Finally, we extracted 500-bp upstream from the start codon of each ortholog and searched for the Mcm1 DNA-binding motif in each upstream putative control region using the AME function of the MEME suite.

## Mcm1 replacement assay

We used CRISPR-Cas9 editing to directly replace the MADS-box domain of Mcm1 in *S. cerevisiae* (strain yMAH302). We edited Mcm1 MADS-box with a sgRNA (5′-AACGACTAGCAACAGGA CCT-3′) targeting a Pam site overlapping codons 56 and 57, and the repair was directed using a dsDNA donor encoding the MADS-box domain of *Huva*Mcm1. Edited colonies were screened by diagnostic PCR/digestions, where the PCR-amplified Mcm1 MADS-box fragment from successfully edited clones was only positively digested with KpnI. Successfully edited clones were Sanger sequenced to confirm the edit. In addition, we isolated WT-edited clones, which only carried the sgRNA abolishing synonymous mutations, and used these as our control strains in the RNAseq experiment, ensuring any bias introduced via CRISPR-Cas9 cloning was correctly controlled. Oligos used are listed in Supplementary Table 3.

We then grew strains to mid-log phase (A600 ~0.6–0.8) and extracted RNA as previously reported (Haase, Ólafsson, *et al.* 2023). We prepared total RNA with rRNA-depletion sequencing libraries using the QIAseq Stranded Total RNA kit (Qiagen Cat. 180745) and the QIAseq FastSelect-rRNA Yeast Kit (Qiagen Cat. 334217). Lastly, libraries were sequenced with an Illumina NextSeq 500 with paired-end 2 × 75-bp read chemistry, generating >22 million reads per sample. Lastly, transcript abundances were quantified from the RNAseq data set with the program kallisto (v.0.46) and the differential expression analysis was performed using the companion program sleuth (v0.30) (Bray *et al.* 2016; Pimentel *et al.* 2017). The *S. cerevisiae* S288C genome build R64-2-1 was used for analysis.

## Flow cytometry and HU arrests

Cells were grown overnight in YPD at 30°C, and the following morning, saturated cultures were diluted to $A_{600} \asymp 0.2$ and grown until mid-log phase, $A_{600} \asymp 0.6$. Cells were then washed in PBS, resuspended in YPD + 300-mM hydroxyurea, and placed at room temperature for 60 minutes with agitation. Arrested cells were then washed 2× in fresh YPD and then resuspended in YPD, and placed at 30°C for outgrowth after HU arrests. We took aliquots of the cell suspension at various timepoints for analysis. For DNA content analysis, cells were first crosslinked with 0.5% paraformaldehyde for 15 minutes at 4°C. After 2× washes with PBS, crosslinked cells were then resuspended in ice-cold (−20°C) 70% methanol and incubated at 4°C for 1 hour. Cells were then washed with 2× PBS, resuspended in PBS + 2.5-µM SytoxGreen, and incubated at 30°C for 30 minutes. For analysis of HTA-mNeonGreen, aliquots of cells were taken at the appropriate timepoints, washed 2× in ice-cold PBS, and placed on ice until analyzed for flow cytometry. Cells were then analyzed using a spectral cell analyzer (Sony SA3800), and data from approximately ~30,000 events were analyzed in the FlowJo software.

## Time-lapse imaging of *H. uvarum*

Prior to imaging cells were grown to mid-log phase ($A_{600} \asymp 0.6$–0.8) in YPD. Cells were then collected, resuspended in SC medium, and placed at room temperature for 1 h. Meanwhile, we prepared a 15 µ-slide VI (ibidi Cat. 80606) for imaging by coating the surface with Concanavalin A from *Canavalia ensiformis* (10 mg/mL in water). Cells were then loaded onto the slide and incubated for 10 minutes prior to 2 washing steps with SC media. Finally, cells were placed into a temperature-controlled EVOS M7000 imaging system, and time-lapses were collected at 30°C with images taken at either 2.5- or 5-minute intervals. For time-lapses after HU arrests, cells were arrested with HU as above. After HU arrest, cells were quickly washed in PBS and resuspended in SC medium, and immediately placed into the imaging chamber. Time-lapses were then acquired the same as above. Movies were then analyzed in Fiji using the TrackMate plugin.

## Results

### Histone gene evolution in a selection of Saccharomycotina yeasts

To better understand the origins of the duplicated histone gene clusters, we searched the genomes of a selection of Saccharomycotina yeasts. The majority of species examined encoded 2 paralogous H2A–H2B and H3–H4 histone gene clusters—similar to *S. cerevisiae* (Fig. 1c). However, we observed that was not the case for early diverging lineages. For example, species within the Lipomycetaceae (*Lipomyces* spp.) and Trigonopsidaceae (*Tortispora* spp.), which diverged >300 MYA (Shen *et al.* 2018), encode a single H2A–H2B and H3–H4 histone gene cluster (Fig. 1c). Intriguingly, we observed that species in the Dipodascaceae/ Trichomonascaceae encoded either a single (*Stramerella*, *Yarrowia*) or 2 (*Nadsonia*) copies of each histone cluster. The most parsimonious interpretation, explaining the distribution of histone gene cluster copy number in extant species, is that the each of the histone gene clusters duplicated twice (once in the ancestor of *Nadsonia* and once in the *Alloascoidea*–*Saccharomyces* ancestor) following the divergence of the Lipomycetaceae and Trigonopsidaceae lineages. Moreover, we observed that the Lipomycetaceae (*Lipomyces* spp.) histone genes were interspaced with introns, an observation not previously noted, whereas the majority of other examined species'

histone gene clusters did not (Fig. 1c; Yun and Nishida 2011). This is consistent with what is known about genome evolution in the Saccharomycotina, where the majority of genes have lost their introns (Goffeau *et al.* 1996).

## Paralogous core histone gene loss and divergence in *Hanseniaspora* FEL

Mapping of histone gene cluster copy number to the phylogeny suggested a case of secondary loss in the genus *Hanseniaspora* (Fig. 2a). In the *Hanseniaspora* SEL, we observed the presence of the normal copy number of histone gene clusters, as in *S. cerevisiae* (Fig. 2a, Supplementary Fig. 1). However, for species of the *Hanseniaspora* FEL, we observed only a single copy of each histone gene cluster (Fig. 2a). Synteny analysis suggests that the histone clusters HTA2B2 and HHF1T1 were lost in the *Hanseniaspora* FEL ancestor (Supplementary Fig. 1b and c). Intriguingly, we also observed a convergent partial loss event in 1 species of the *Hanseniaspora* SEL, *H. gamundiae*, which lost paralog cluster HHT1F1, suggesting that it may represent an independent intermediate state or an artifact of incomplete genome sequencing and assembly (Fig. 2a, Supplementary Fig. 1a).

## *H. uvarum* H2A–H2B histone dimer is functionally divergent and incompatible with *S. cerevisiae*

Protein alignments showed that *Hanseniaspora* FEL histones diverged from the SEL and other yeasts, likely owing to the well-known rapid burst of evolution in the stem of the FEL (Supplementary Fig. 2; Steenwyk *et al.* 2019). We used a histone replacement assay in *S. cerevisiae* to examine the functional significance histone divergence in the *Hanseniaspora* FEL (Haase *et al.* 2019). In this scheme, we used a *S. cerevisiae* strain in which the native histone clusters are deleted from their chromosomal loci and a single set of core histone genes (HTA2B2–HHF1T1) is provided on a counter-selectable "Superloser" plasmid (Haase *et al.* 2019). Using the plasmid shuffling method, the native core histone genes are readily eliminated and swapped for an incoming set of heterologous histone genes (Fig. 2b). As histones H2A and H3 were the most incompatible between yeast and human (Truong and Boeke 2017), we first exchanged individual *Hanseniaspora* species' H2A and H3 histones and found that these 2 individual histones readily functioned in *S. cerevisiae*, though, histones from FEL species generally performed worse than those from SEL species (Fig. 2c–e).

Given that the individual histones swapped well, we next attempted to swap in all 4 of *H. uvarum*'s histone genes at once. However, *H. uvarum*'s histone genes, under the control of *S. cerevisiae* histone control regions, did not complement in *S. cerevisiae* (Fig. 2f). As a positive control, we swapped all 4 histones from the closely related species *Saccharomyces eubayanus*, which complemented well in our histone shuffle assay (Fig. 2f; Haase *et al.* 2023b). To determine which of *H. uvarum*'s histones are inviable in *S. cerevisiae*, we first individually replaced each of *H. uvarum*'s histones with the homolog from *S. cerevisiae*. We observed weak complementation when we replaced either HuvaH2A or HuvaH2B with ScH2A or ScH2B, respectively (Fig. 2g and h), suggesting that the HuvaH2A–H2B dimer is incompatible with *S. cerevisiae*. We confirmed this by replacing the HuvaH2A–H2B dimer with the ScH2A–H2B dimer (in the context of the HuvaH3–H4), which resulted in full complementation (Fig. 2i).

While the individual histone swaps (H2A and H3) worked well (Fig. 2c–e), the double swap of HuvaH2A–H2B failed to compliment entirely (Fig. 2i). Moreover, the triple swaps of HuvaH2A–H3–H4 or HuvaH2B–H3–H4 only barely complement and gave phenotypical small colonies (Fig. 2g and h). These observations suggest that

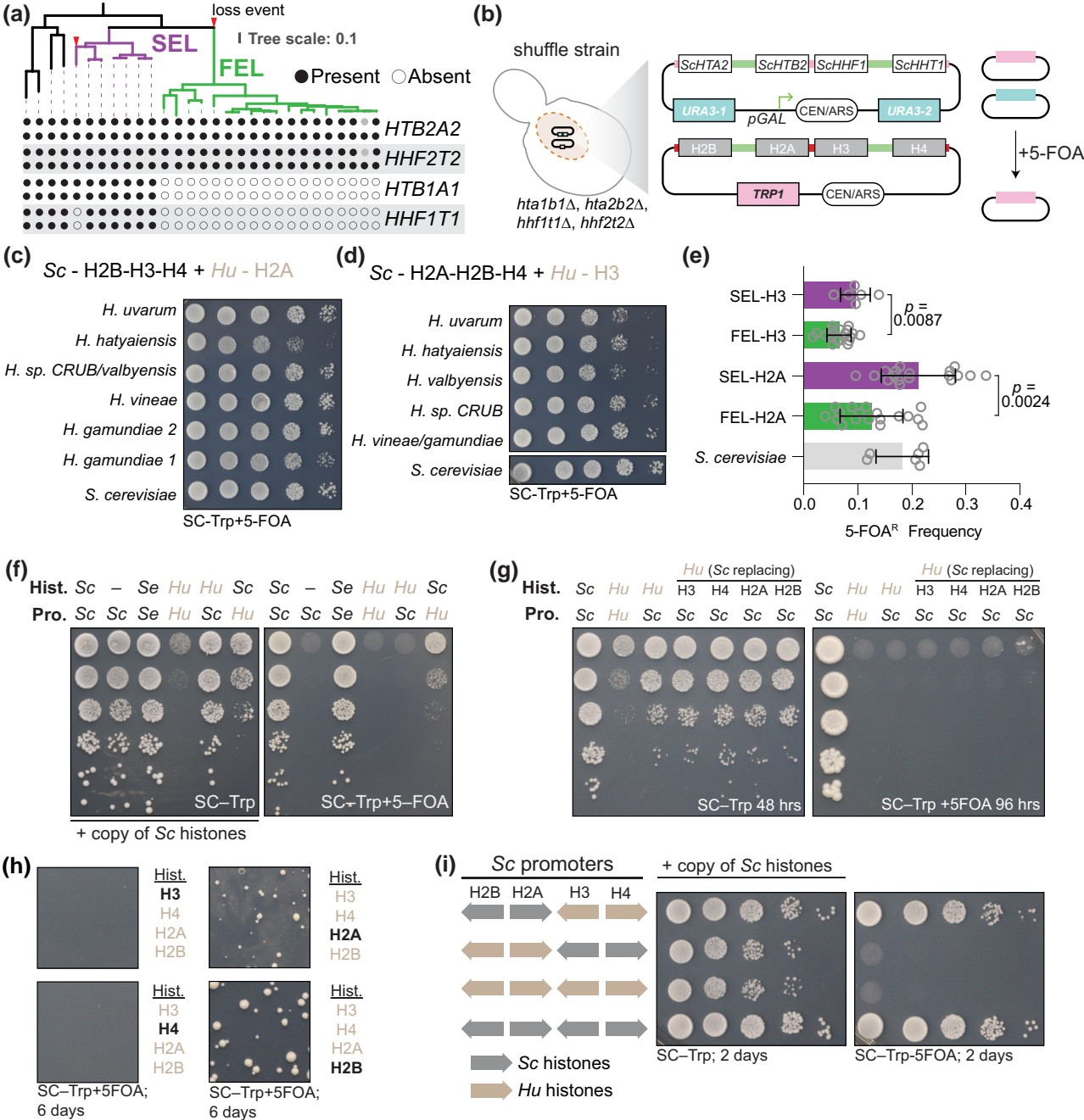

**Fig. 2.** Paralogous gene loss and divergence of core histones in *Hanseniaspora* FEL. a) Phylogeny of the *Hanseniaspora* and 4 outgroup species from Steenwyk *et al.* (2019), showing the presence and absence of the core histone gene clusters. Purple, the slow-evolving lineage; green, the fast-evolving lineage. Outgroup lineages from left to right: *S. cerevisiae*, *K. lactis*, *Cyberlindnera jadinii*, *Wickerhamomyces anomalus*. The full phylogeny, complete with species and strain names, can be found in Supplementary Fig. 1a. b) Overview of dual-plasmid histone shuffle assay in *S. cerevisiae*. Details of plasmid shuffling can be found in the *Materials and methods* and also in Haase *et al.* (2019, 2023a). c) Histone swaps of H2A orthologs from SEL (*H. gamundiae, H. vineae*) and FEL (*H. sp. CRUB/valbensis, H. hatyaiensis, H. uvarum*) species. d) Histone swaps of H3 orthologs from SEL (*H. gamundiae* and *H. vineae*) and FEL (*H. sp. CRUB 1602/ valbyensis, H. hatyaiensis, H. uvarum*) species. Placement of 2 species names on either side of "/" indicates their histone amino acid sequence was the same. e) Quantifications of 5-FOA$^R$ frequency from panels c) and d), see *Materials and methods* for details. Each histone was shuffled in n = 6 biological replicates and results were aggregated based on SEL/FEL classification. Statistical significance in difference of the median 5-FOA$^R$ frequency between SEL/FEL histones was determined by Mann–Whitney test. f) Shuffle assay of *H. uvarum*'s histones and histone control regions in *S. cerevisiae*. Left is the growth assay maintaining the selection for both plasmids, and right is the counterselection (5-FOA) of the native *S. cerevisiae* histone plasmid. Abbreviations are used for displaying which host species each histone control regions and genes were sourced from Sc—*S. cerevisiae*; Se—*Saccharomyces eubayanus*; Hu— *H. uvarum*. g and h) Histone swaps of *H. uvarum* core histone genes and with individual replacements with *S. cerevisiae* histones. g) To identify which of the 4 histones from *H. uvarum* were inviable in *S. cerevisiae*, we replaced each single *H. uvarum* histone with its ortholog from *S. cerevisiae*. Only when we replaced either *Huva*H2A or *Huva*H2B did we observe weak complementation, indicating that the H2A–H2B dimer combination is not viable. h) The number of cells for each complementation was scaled up (~10$^8$ cells) and plated onto an entire 10-cm petri dish; the bolded histone in black coloring indicates the *H. uvarum* histone that was replaced with the *S. cerevisiae* homolog on the incoming histone plasmid. i) Shuffle assay of *H. uvarum*'s H2A–H2B and H3–H4 in *S. cerevisiae*. Selection and counterselection are the same as in panels c) and d). Histones in panels h) and i) were all expressed under the native *S. cerevisiae* histone control regions.

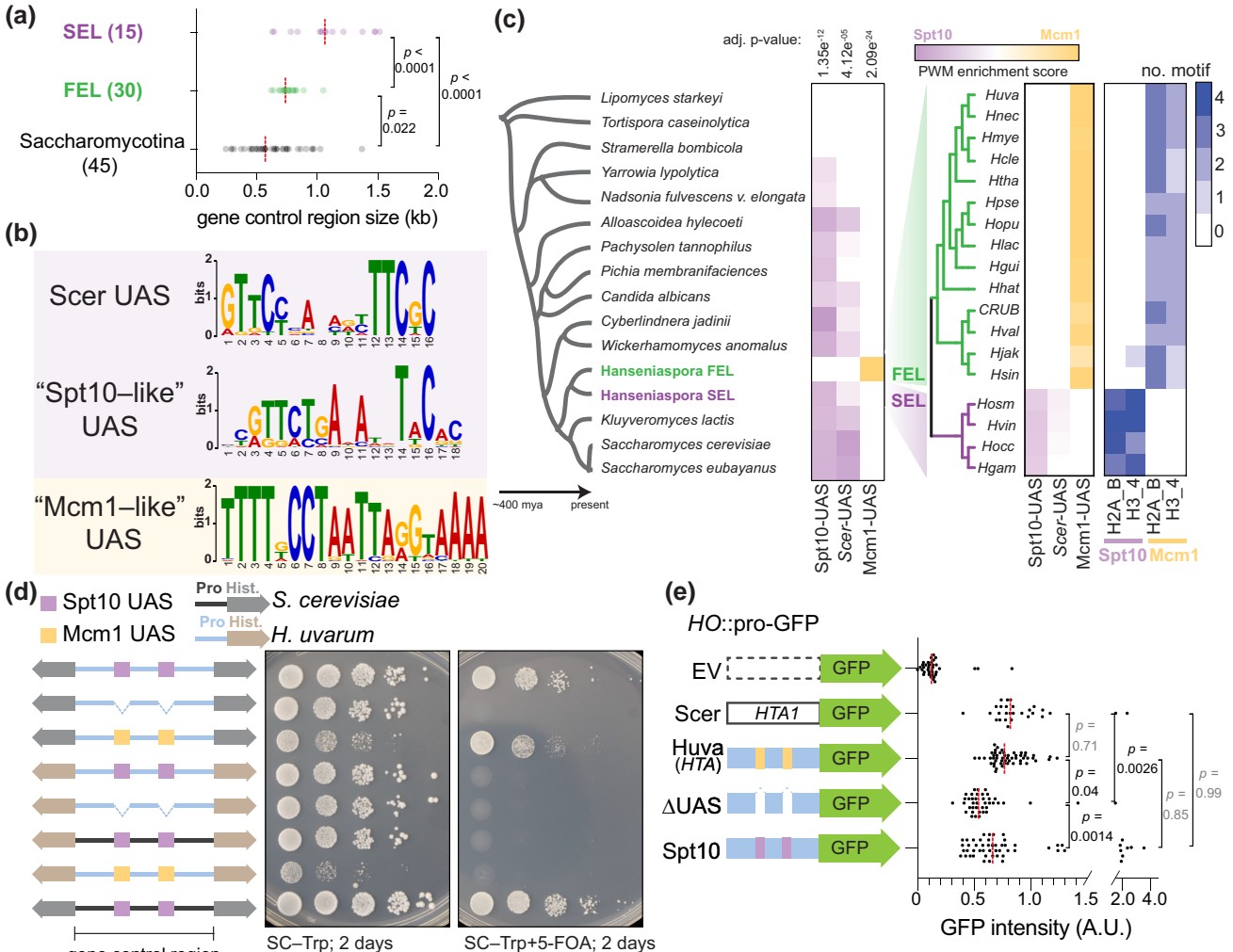

**Fig. 3.** Core histone cis-regulatory rewiring in *Hanseniaspora* FEL. a) Histone control regions sizes in outgroup Saccharomycotina species (species shown in Fig. 1a) and the *Hanseniaspora* SEL and FEL species. The number of histone control regions examined for each is shown. b) Motifs discovered (MEME) from a search of histone control regions from *Hanseniaspora* and 4 outgroup species (*S. cerevisiae*, *K. lactis*, *C. jadinii*, *W. anomalus*). The "Scer UAS" (Scer Upstream Activating Sequence) motif was constructed using known Spt10 DNA-binding sites of the 4 core histone genes (Eriksson *et al.* 2012). c) Histone control region motif enrichment analysis of the Spt10 and Mcm1 DNA-binding sites. Histone control regions were searched for enrichment of either *S. cerevisiae*'s Spt10 motif, the conserved Spt10 motif found across taxa ("Spt10-like" UAS), and the Mcm1 motif found in *Hanseniaspora* FEL ("Mcm1-like" UAS) using the AME motif enrichment (MEME suite). The average position weight matrix (PWM) of all identified motifs from each species is shown, with a dual–color scheme showing enrichment for Spt10 or Mcm1 (purple to yellow). Lastly, the number of identified motifs is shown. d) Functional dissection of the Mcm1 binding sites in histone control regions of *H. uvarum*. Plasmid shuffle assay was carried out as in Fig. 1b. Shown to the left of the growth assays are diagrammatic representations of the constructs tested (grey histones = *S. cerevisiae*; brown histones = *H. uvarum*; black histone control region = *S. cerevisiae*; blue histone control region = *H. uvarum*; purple box = Spt10 UAS; yellow box = Mcm1 UAS). All 4 histones were swapped in all cases, but only 1 histone gene cluster is shown for simplicity (for *Huva*H2A–H2B, 3 Mcm1 sites are present, and for *Huva*H3–H4, 2 sites are present). Colored links between gene arrows represent the species' histone control region used, whereas colored boxes represent the specific DNA-binding element present at the UAS sites. Additionally, the species' histones are color-coded, as shown. e) Expression analysis of a GFP reporter targeting the *HO* locus with no upstream histone control region or the indicated histone control region (all *HTA1B1* histone control regions). Total GFP fluorescence was measured from 10 images acquired using an EVOS M7000 imaging system. Statistical significance of the mean difference in GFP fluorescence was determined with a 1-way ANOVA test and corrected for multiple comparisons with hypothesis testing (Šídák).

epistatic interactions between *Huva*H2A and *Huva*H2B dominate the incompatibility of *H. uvarum*'s histones in *S. cerevisiae*, alongside contributions from interactions between either *Huva*H2A or *Huva*H2B and *Huva*H3–H4 tetramer. These data are correlated to the divergence of each histone, where *Huva*H2B and *Huva*H2A show more numerous amino acid substitutions than compared to *Huva*H3 and *Huva*H4. Interestingly, this contrasts a previous report of human histone complementation in *S. cerevisiae*, where human H2A–H2B complemented significantly better than human H3–H4 (Truong and Boeke 2017). Moreover, the potent genetic

suppressor of human histones in yeasts, *DAD1*[E50D] (Haase, Ólafsson, *et al.* 2023), did not rescue the inviability of *H. uvarum*'s H2A–H2B dimer (data not shown), suggesting that there may be species-specific incompatibilities at play.

## Core histone gene cis-regulatory innovation in *Hanseniaspora* FEL

A striking dominant negative effect was observed when the histone control regions of *H. uvarum* (*Huva*HCR—the intervening DNA sequence between the 2 bidirectional transcribed histone

genes) were used to express either *H. uvarum* or *S. cerevisiae* histones (Fig. 2f; column 4 or 6). We probed whether underlying sequence differences in these histone control regions led to decreased fitness. To this end, we examined the histone control regions from a set of *Hanseniaspora* and outgroup species (see Materials and methods) and tested for enrichment of DNA sequence motifs. The histone control regions varied in size, with FEL species and other Saccharomycotina yeasts having markedly shorter histone control regions than the SEL species (Fig. 3a). We identified a motif corresponding to the DNA-binding sequence of the conserved core histone gene regulator Spt10 in most outgroup species and throughout the SEL lineage (Fig. 3b). However, the Spt10 motif was notably absent from species in the *Hanseniaspora* FEL lineage. From a discriminative de novo motif search, we identified a second motif corresponding to the DNA-binding sequence of the transcription factor Mcm1 (Fig. 3b). Mapping the presence of the Mcm1-like motif to the phylogeny showed that it was exclusive to histone control regions of the FEL lineage (Fig. 3c)—suggesting that the FEL lineage underwent an ancestral histone gene regulatory rewiring event and that perhaps these sites may be responsible for the *Huva*HCR toxicity in *S. cerevisiae*. We also identified a second motif that was enriched in the FEL species' histone control regions corresponding to the Rap1 DNA-binding site (Supplementary Fig. 3a and b). However, this site was not found across all species in the FEL and was notably absent from the majority of the histone H3–H4 control regions (Supplementary Fig. 3c); as such, we did not investigate the putative Rap1 site further.

## Mcm1 DNA-binding sites underlie toxicity of *H. uvarum* histone control regions in *S. cerevisiae*

To determine whether the Mcm1 DNA-binding sites were responsible for the growth defect, we generated mutants of the 2 *Huva*HCRs (here, *Huva*HCRs refer to the HCR of the H2A–H2B and H3–H4 gene clusters) with the Mcm1 UASs removed (*Huva*HCR$^{\Delta Mcm1}$). Deletion of Mcm1 UAS resulted in amelioration of the toxic phenotype, and, as expected, the *Huva*HCR$^{\Delta Mcm1}$ was no longer viable when expressing *S. cerevisiae* histones (Fig. 3d). We restored viability to the *Huva*HCR$^{\Delta Mcm1}$ construct by replacing the Mcm1 UAS for the consensus Spt10 binding motif from the UAS of the histone control regions of *S. cerevisiae* (*Huva*HCR$^{UAS-Spt10}$). Remarkably, the *Huva*HCR$^{UAS-Spt10}$ was no longer toxic when driving the expression of histones (Fig. 3d, see SC–Trp), and, importantly, *Huva*HCR$^{UAS-Spt10}$ was sufficient for viability when expressing the histones of *S. cerevisiae* (Fig. 3d, see SC–Trp+5-FOA). Interestingly, deletion of the Mcm1 paralog, Arg80, sharpened the toxicity of *Huva*HCR (Supplementary Fig. 4a), consistent with the idea that removal of competitive binding by Arg80 may allow for increase binding of Mcm1 to *Huva*HCR, potentiating its toxic effects. We therefore conclude that the Mcm1 sites are the functional elements responsible for the toxicity of *Huva*HCR in *S. cerevisiae*.

The toxicity of the *Huva*HCR could be due to histone overexpression, temporal misexpression, or both. We used a GFP reporter assay to investigate the level of activity from the histone control regions of H2A genes. We observed that the *Huva*HCR-H2A showed similar levels of GFP intensity to the native *Sc*HCR-H2A (Fig. 3e; the HTA1 histone control region). Moreover, removing the Mcm1 UAS sites in *Huva*HCR-H2A reduced the expression of GFP and the insertion of the Spt10 UAS restored expression to normal levels (Fig. 3e). These data support the idea that the *H. uvarum* histone control regions do not lead to histone overexpression per se but perhaps temporal misexpression. Although we did not formally test this idea, it

potentially explains the striking dominant negative growth effect in *S. cerevisiae*, as expression of histones outside of S-phase has a well-known cytotoxic effect (Kurat, Recht, *et al.* 2014). In conclusion, we show that the Mcm1-mode is incompatible in species with the ancestral Spt10-mode of histone gene regulation.

## *H. uvarum* Mcm1 functionally replaces *S. cerevisiae* Mcm1

The above data strongly indicate that Mcm1 regulates the core histones in *Hanseniaspora* FEL. To gain insight into whether the core histones of the *Hanseniaspora* FEL shifted into a novel regulation paradigm (Mcm1-mode) or if Mcm1 functionally diverged during the evolution of *Hanseniaspora* FEL, we performed 2 tests. First, we explored the evolution of cis-regulatory sites of Mcm1 target genes in *H. uvarum*, by performing a motif search of the Mcm1 DNA-binding sequence in the control regions of orthologs of *Sc*Mcm1-regulated genes. We found that the large majority of orthologs of *Sc*Mcm1-regulated genes in *H. uvarum* also have Mcm1 binding sites in their putative control regions (Supplementary Fig. 4b), suggesting that *Huva*Mcm1 regulates the same set of target genes in *H. uvarum* as *Scer*Mcm1 does in *S. cerevisiae*.

Next, we tested *Huva*Mcm1's function in *S. cerevisiae* by examining whether *Huva*Mcm1 could complement *Scer*Mcm1 in *S. cerevisiae* (Fig. 4a). Specifically, we tested for the essential function of *Scer*Mcm1, which is sufficiently conferred by its DNA-binding MADS-box domain (Fig. 4a, Supplementary Fig. 5a). To this end, we replaced the MADS-box domain of *S. cerevisiae*'s Mcm1 (*Scer*Mcm1) with the orthologous Mcm1 MADS-box domain from *H. uvarum* (*Huva*Mcm1). Scarless *Scer*Mcm1::*Huva*Mcm1 MADS-box domain replacements were generated via CRISPR-Cas9 genome editing at the native *MCM1* locus and the successful isolation of edited clones indicated that *Huva*Mcm1 retains the essential functions of *Scer*Mcm1 (Supplementary Fig. 5b and c). Growth assays revealed that these strains showed no phenotypic difference, as *Huva*Mcm1 grew identically to *Scer*Mcm1 cells (Fig. 4b). Moreover, assessment of the transcriptomic effects of *Huva*Mcm1 by RNA sequencing showed that *Huva*Mcm1 had little effect on *S. cerevisiae*'s transcriptome, with only 5 genes being significantly dysregulated (Fig. 4c; Supplementary Fig. 5d–g). These data support the conclusion that *Huva*Mcm1 is conserved in function—both its DNA-binding specificity and its target genes—with *Scer*Mcm1. Conservation of function of *Huva*Mcm1 is consistent with a model in which the core histones were rewired into a preexisting regulatory network rather than *Huva*Mcm1 diverging in function in *Hanseniaspora* FEL.

## Rapid cell division and decoupled core histone and DNA synthesis *in H. uvarum*

Given the cis-regulatory divergence of core histone genes, we were curious whether the dynamics of core histone expression are altered in *H. uvarum*. Using a recently described method for the genetic manipulation of *H. uvarum*, we inserted an H2A-mNeonGreen (a LanYFP-derived fluorophore; Shaner *et al.* 2013) fusion construct at the native *HTA1* locus (HTA-mNG; Fig. 5a and b). Similar systems have been used to monitor cell cycle dynamics and histone protein synthesis in *S. cerevisiae* (Garmendia-Torres *et al.* 2018). We tracked the nuclear intensity of H2A over the course of a few cell cycles (Fig. 5c–f), measuring an average cell cycle length of ~60 minutes for cells grown in SC medium without exposure to the excitation laser (Supplementary Fig. 6a; Supplementary Video 1). In contrast, when grown with exposure every 5 minutes, cells had a slightly increased cell cycle length

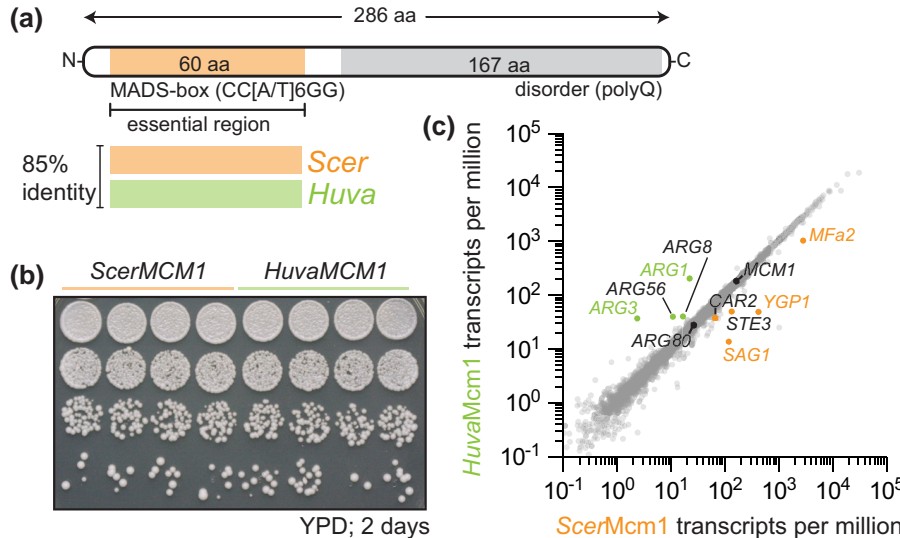

**Fig. 4.** Mcm1 essential function and ancestral regulatory network are conserved. a) Schematic of the Mcm1 protein, highlighting the MADS-box (MCM1, AGAMOUS, DEFICIENS, SRF) domain, Mcm1's essential DNA-binding domain (Messenguy and Dubois 2003), and conservation between *S. cerevisiae* (*Scer*Mcm1; orange) and *H. uvarum* (*Huva*Mcm1; green). The MADS-box domain of Mcm1 from *H. uvarum* was used to directly replace *S. cerevisiae*'s Mcm1 MADS-box domain. b and c) *Huva*Mcm1 has little effect on the *S. cerevisiae* transcriptome. b) Fitness assay of WT and the HuvaMcm1 *S. cerevisiae* strains. Strains carrying *Scer*Mcm1 or *Huva*Mcm1 were grown overnight in YPD medium; the following day, cultures were normalized ($A_{600} \simeq 1.0$), and cells were spotted on agar plates and grown for 2 days at 30°C. n = 4 biological replicates. c) Transcriptomic effects of *Huva*Mcm1. X–Y plot of transcript abundance in *Scer*Mcm1 vs *Huva*Mcm1 strains. The average transcript per million from 4 biological replicates is plotted. Genes that were observed to be differentially expressed are labeled—with statistically significant differentially expressed genes indicated by colored labels (either labeled green or orange; Supplementary Fig. 5f, Supplementary Table 4). The expression levels of Mcm1 and Arg80 are labeled. See *Discussion* for details on expression changes.

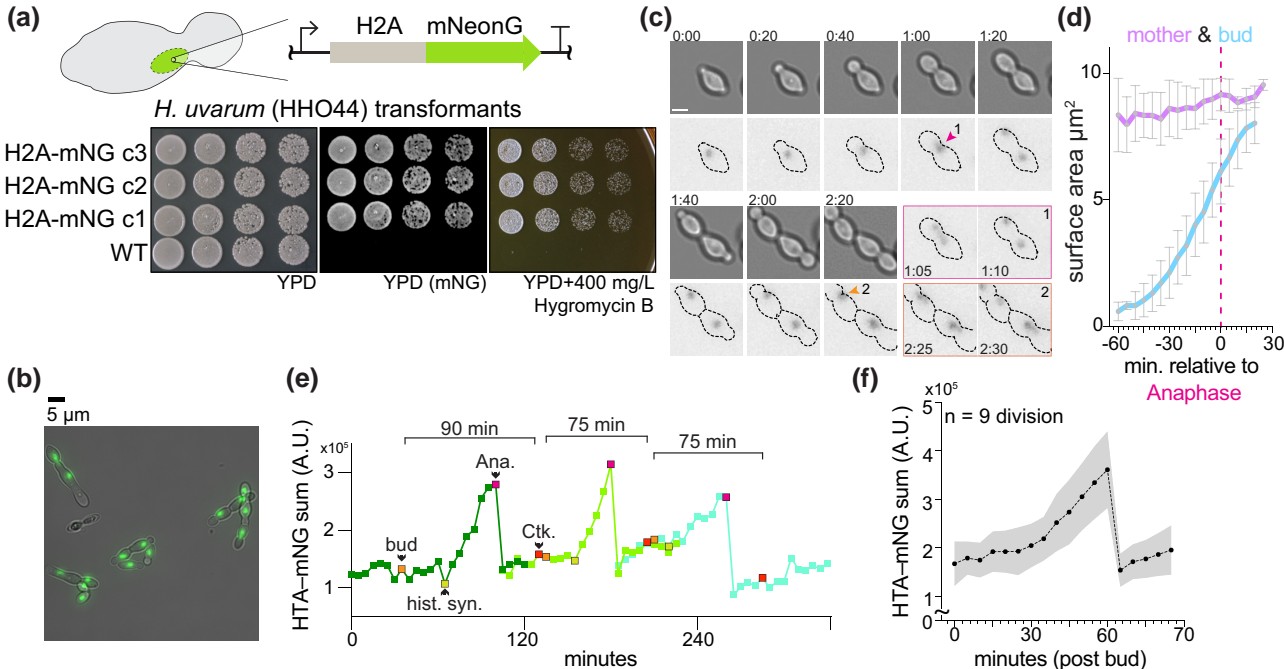

**Fig. 5.** Cell cycle dynamics and histone synthesis in *H. uvarum*. a) Schematic of H2A-mNeonGreen-hphMX cassette used to target the endogenous H2A gene. Note that the hphMX cassette used for hygromycin B resistance is not depicted but is immediately downstream of the terminator of the H2A-mNG fusion. Below is a spot assay of 3 transformed clones grown on YPD (imaged on a Bio-Rad ChemiDoc MP imaging system with blue epi-illumination) and YPD with 400-mg/L hygromycin B. b) Example image of *H. uvarum* cells with H2A-mNG tag. Scale bar = 5 μm. c) Time-lapse growth of *H. uvarum* with H2A-mNG tag. A series of images are shown from Supplementary Video 2 at intervals of 20 minutes. Two divisions are separately shown in the images outlined in magenta and orange. Scale bar = 2.5 μm; time H:MM. d) Mother and daughter (bud) cell surface area analysis. The surface area was determined by manually outlining mother and daughter pairs (n = 5). Mean areas with ± SD are shown. The bud grew ~8-fold faster than the mother cell; mother = $0.84 \pm 0.24$ μm$^2$ hour$^{-1}$; bud = $6.6 \pm 0.3$ μm$^2$ hour$^{-1}$. e) Example track of H2A-mNG levels from time-lapse growth of *H. uvarum* cells. Three divisions are followed, an orange-colored point indicates bud emergence (bud); a yellow point indicates the start of histone synthesis (hist. syn.); a magenta point indicates the start of mitosis/anaphase (Ana.); a red point indicates the completion of division (cytokinesis; Ck.). Time-lapse images were acquired every 5 minutes at 30°C in SC medium. f) Average H2A-mNG levels during the cell cycle of *H. uvarum*. Movies from 9 cells were all set relative to bud emergence (t = 0), and H2A-mNG levels were tracked and quantified until cytokinesis.

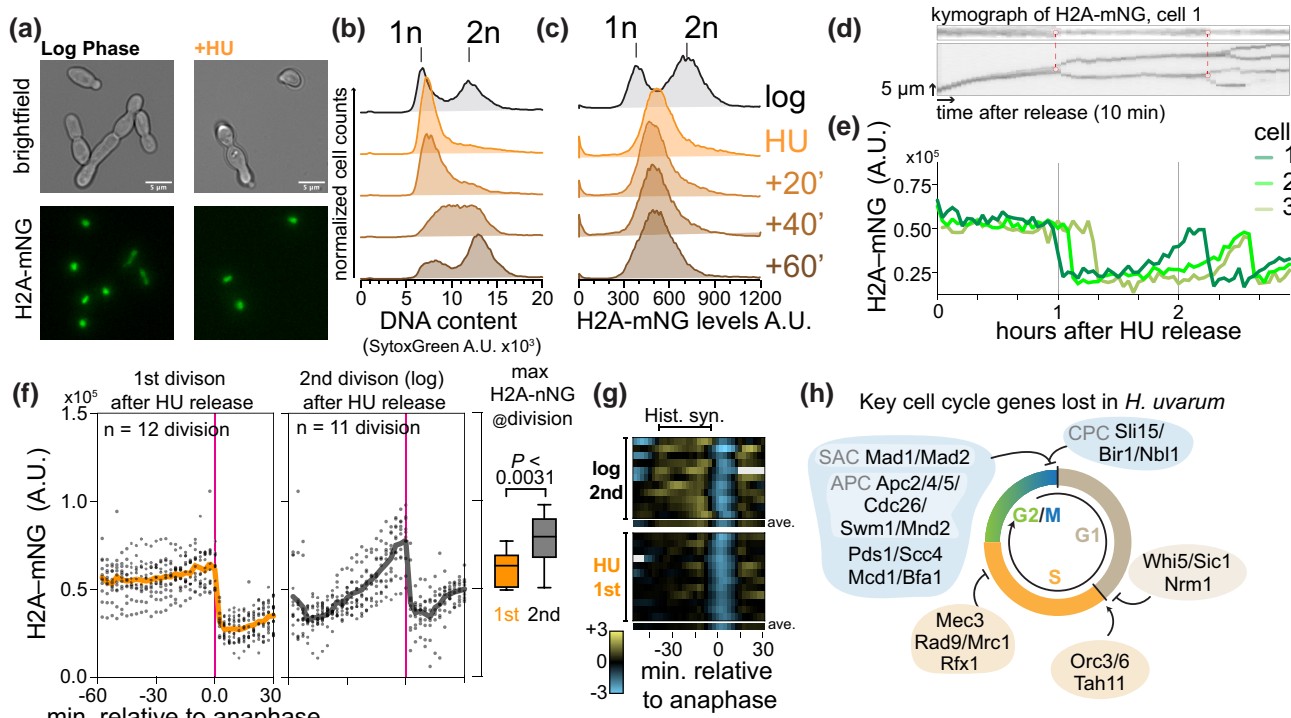

**Fig. 6.** Core histone synthesis is decoupled from DNA synthesis in *H. uvarum*. a) Example images of cells from mid-log phase growth or after 60 minutes of arrest in 300 mM HU. b and c) DNA content analysis and histone abundance (H2A-mNG level) by flow cytometry of cells from mid-log phase growth or after, and following release, from 60 minutes of early S-phase arrest in HU. d and e) Histone levels do not appreciably increase in single cells following release from HU. d) Kymographs of H2A-mNG levels from cell 1 (the top kymograph is focused on the nucleus of the mother cell, and the bottom is a kymograph focused on a region containing the nuclei of mother and daughter cells), connected red dots indicating the time of division, and dashed lines denote the 2 divisions tracked below. The arrow representing time corresponds to 10 minutes, and the arrow representing space corresponds to 5 μm. e) Example tracks of the sum of H2A-mNG levels (arbitrary units; A.U.) in 3 mother cells monitored for up to ∼3 hours post-release from HU. f) Histone level dynamics in the first and second cell divisions after HU arrest. H2A-mNG levels were quantified relative to anaphase (magenta line, timepoint = 0), at which time histone levels remained constant prior to the first division after release from HU. In contrast, histone levels showed a characteristic linear increase during the second division after HU release. Average profiles are shown as solid lines. To the right is the maximum H2A-mNG signal just prior to anaphase, showing that histone levels were higher during the second cell division after HU release. g) Rate of histone synthesis (H2A-mNG) in the first division after HU arrest (n = 12) or during the second round of division (n = 11). The first derivative of H2A-mNG levels is plotted for each replicate cell (log scale). Positive values (yellow) correspond to instantaneous increases in histone levels (synthesis), no change corresponds to zero (black), and negative values (blue) correspond to instantaneous decreases in histone levels (anaphase). Missing values are filled in as gray rectangles. Each rectangle represents a 5-minute interval. Below each condition, the average profile of that condition is plotted. h) Overview of key cell cycle regulators lost in *H. uvarum*. Data from Steenwyk *et al.* 2019. We hypothesize that a common signal initiates DNA replication and histone synthesis in *H. uvarum*; however, due to the loss of negative regulation (loss of *HPC2*) and the shift to the Mcm1-mode regulation, core histone synthesis is no longer coupled to the status of DNA replication. As various checkpoints during the cell cycle have been lost, it may be advantageous to fully commit to histone production even if DNA replication is perturbed. Thus, while gene loss and cis-regulatory rewiring potentially led to the loss of the DNA synthesis-dependent regulation of histone synthesis, *H. uvarum* ensures that histones are produced in a timely manner consistent with its rapid progression through the cell cycle.

of ∼80 minutes, which we observed in either WT or fused H2A-mNG cells, suggesting a slight phototoxic effect (Supplementary Fig. 6a; Supplementary Video 2). By examining the divisions of single *H. uvarum* cells, we observed that their cell cycle dynamics are markedly different than in *S. cerevisiae* in several ways (for comparative data from *S. cerevisiae*, we refer the reader to Garmendia-Torres *et al.* 2018). First, both daughter and mother cells synchronously bud following cytokinesis (Fig. 5c; Supplementary Video 1), whereas in *S. cerevisiae*, daughter cells display prolonged G1 and S phases. Second, histone synthesis begins well after bud emergence (∼30 minutes; Fig. 5e and f), in contrast to *S. cerevisiae*, where histone synthesis begins just prior to bud emergence. Third, the nascent bud reaches near–full mature cell size just prior to mother–daughter cytokinesis (Fig. 5d; Supplementary Video 2); however, in *S. cerevisiae*, the nascent bud does not reach the size of the mother until later cell cycles. Taken together, we show that alongside their rapid cell cycle progression, histone synthesis and bud growth are significantly different in *H. uvarum* in comparison to *S. cerevisiae*.

In most eukaryotes, histone protein synthesis is tightly coupled to the status of DNA replication (Eriksson *et al.* 2012; Rattray and Müller 2012; Kurat, Recht, *et al.* 2014). In *S. cerevisiae*, histone synthesis is inhibited following a DNA replication block with the treatment of hydroxyurea (HU) via a specific regulatory coupling (Bhagwat *et al.* 2021), which is likely mediated by the HIR complex [histone regulatory genes (*HIR1*, *HIR2*, *HIR3*) and histone periodic control gene (*HPC2*)]. Given the altered cell cycle dynamics, we were curious whether histone synthesis was dependent on DNA synthesis in *H. uvarum*. We arrested cells in early S-phase using HU and followed cell division after release (Supplementary Video 3). We observed that after 60 minutes of HU arrest, histone levels were near expected mitotic levels (Fig. 6a–e), whereas the DNA remained unreplicated, as confirmed by DNA content analysis by flow cytometry (Fig. 6b). Following HU release, the cells resumed growth and completed DNA synthesis within ∼60 minutes (Fig. 6b); however, during this same period, histone levels remained constant (Fig. 6c). We next tracked H2A-mNG levels in single cells following HU release. We confirmed that histone levels do

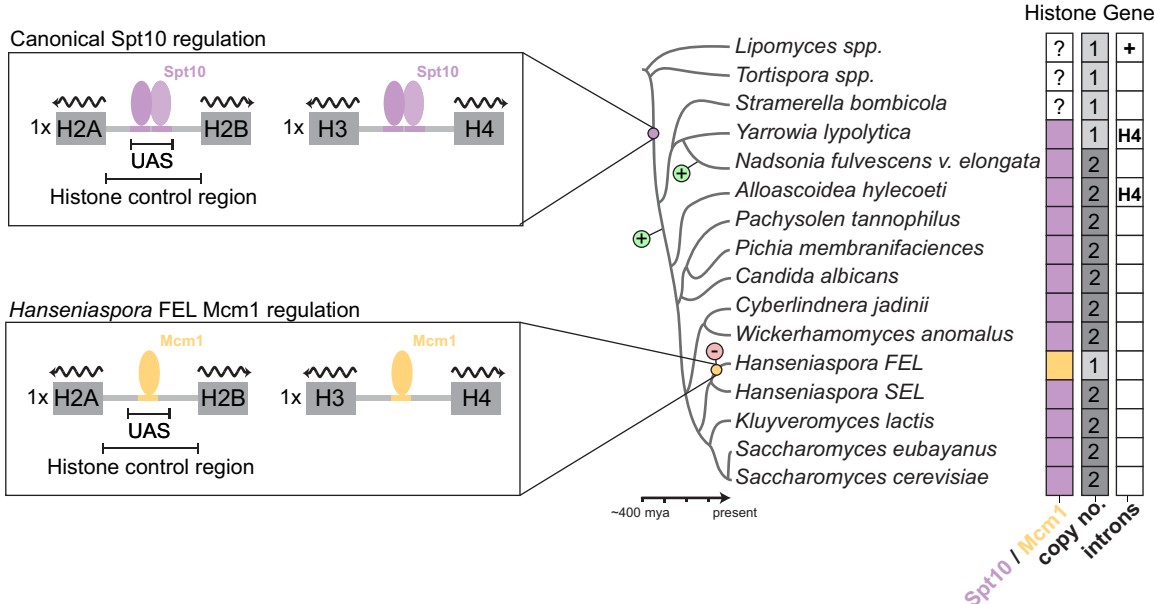

**Fig. 7.** Evolution of core histones in Saccharomycotina. A simplistic model of key events in the evolution of histone gene clusters of Saccharomycotina. The rewiring event from Spt10- to Mcm1-mode in the ancestor of the *Hanseniaspora* FEL is shown. The balloons with a "+" or "−" sign denote the gains or losses of histone gene clusters (both H2A–H2B and H3–H4).

not increase in individual cells during the first cell division after HU release (Fig. 6d–g), with most cells completing division within ~60–120 minutes after HU release (Supplementary Fig. 6b). Moreover, we observed that in the subsequent division, histones were normally synthesized and the total histone levels reached a significantly higher maximal level than during the first division following HU arrest (Fig. 6d–g). Intriguingly, the mother cells divided significantly slower in this second division than their daughter cell (Supplementary Fig. 6c). In sum, histone synthesis in *H. uvarum* is independent of status of DNA synthesis, as histone levels reached mitotic levels during HU arrest and prior to completion of DNA synthesis and did not increase during DNA replication after HU release.

## Discussion

We have detailed the evolution of core histone gene clusters across the budding yeast phylogeny (Fig. 7). From this we are able to infer that the last budding yeast common ancestor (BYCA) had a single copy of each core histone gene cluster, where each histone gene was interspaced introns, and transcriptional regulation was carried out by yet-to-be-identified trans-regulators that bound to its histone control regions. The emergence and fixation of the Spt10 regulatory mode are likely ancient [~320–380 MYA; divergence times from Shen *et al.* (2018) and occurred after the divergence of the Trigonopsidaceae (*Tortispora* spp.), but before the divergence of the Dipodascaceae/Trichomonascaceae (*Nadsonia*, *Yarrowia*, and *Stramerella*)]. Based on the presence and number of histone gene clusters, we predict that the histone gene clusters duplicated twice independently after the emergence of the Spt10 regulatory paradigm. Thus, the contemporary set of core histone gene clusters and regulatory mode (Spt10) emerged after the divergence of the Alloascoideaceae (~250 MYA).

We observed that one set of paralogous core histone genes were lost in *Hanseniaspora*. Gene cluster loss coincided alongside a cis-regulatory rewiring event, swapping the ancestral cis-regulatory Spt10-mode with a derived Mcm1-mode. As both events occurred in the stem of the FEL, we cannot determine the precise ordering of events, although gene loss occurring prior to rewiring would allow for fewer needed cis-regulatory changes for fixation of the Mcm1-mode. Interestingly, the SEL species *H. gamundiae* may represent a case of convergent evolution, but at a more intermediate state, where paralogous histone gene was partially lost and no cis-regulatory changes have yet occurred. In addition to the association of histone gene loss and cis-regulatory rewiring, we also observed that histone gene duplication occurred alongside predicted cis-regulatory change in more basal lineages (e.g. the emergence of the Spt10 regulatory mode). It is tempting to speculate that histone cluster copy number changes are associated with cis-regulatory changes, although the broader significance of such an association remains to be investigated in other lineages.

Moreover, we show that core histone gene regulation is substantially altered in *Hanseniaspora* compared to the model yeast *S. cerevisiae*. Intriguingly, expression of histones under control of the Mcm1-mode in a native Spt10-mode species (*S. cerevisiae*) resulted in a severe growth defect, suggesting that the Mcm1-mode is at least sufficient to illicit phenotypically significant gene expression changes. Changes to gene expression patterns have been observed frequently in the fungal lineage (Gasch *et al.* 2004; Tanay *et al.* 2005; Borneman *et al.* 2007; Tuch, Li, *et al.* 2008; Booth *et al.* 2010; Baker *et al.* 2012). This can occur through changes in the trans-regulator (TR) itself, where alterations of an upstream "Master regulator" may propagate as changes in expression patterns of downstream targets (Tsong *et al.* 2006; Hittinger and Carroll 2007; Pérez *et al.* 2014; Britton *et al.* 2020). In our observed case of the Spt10 to Mcm1 swap of *Hanseniaspora* FEL histone genes, we propose that no changes to the TR Mcm1 occurred. Instead, *Huva*Mcm1 represents a "living ancestor" of the *MCM1* gene prior to the tandem duplication that gave rise to the paralogous gene pair *MCM1*/*ARG80* in the lineage leading to *S. cerevisiae* (Messenguy and Dubois 1993). Indeed, the few transcriptional changes induced by *Huva*Mcm1 were anticipated based on the evolutionary history of Mcm1 in *S. cerevisiae* (Jamai *et al.* 2002; Mead *et al.* 2002; Baker *et al.* 2013). In *S. cerevisiae* a heterodimer of Mcm1 and Arg80, together with the

cofactor Arg81, regulate the *ARG* genes and a homodimer of Mcm1, together with the cofactor Matα1, regulate the α-specific genes. The historical set of mutations acquired by each paralog was result of minimizing paralog interference between the 2 transcription factors due to their cooperative regulation of the *ARG* genes and α-specific genes (Baker *et al.* 2013). Specifically, Arg80 evolved reduced DNA-binding affinity and loss of the Matα1 interaction and Mcm1 lost the Arg81 interaction and stabilized the Matα1 interaction (Jamai *et al.* 2002; Baker *et al.* 2013). Indeed, the downregulation of the α-specific genes is expected since *Huva*Mcm1 lacks the phenylalanine residue at position 48 that is critical for stabilizing *Scer*Mcm1's interaction with Matα1 (Fig. 4c, Supplementary 5a and f; Mead *et al.* 2002). On the other hand, disruption to the regulation of the *ARG* genes [upregulation of Mcm1 repressed (*ARG3*, *ARG1*, *ARG5,6*) and downregulation of Mcm1 activated (*CAR1*, *CAR2*)] is likely due to the effect of "paralog interference" as *Huva*Mcm1 does not have the historical set of amino acid substitutions which minimized Mcm1–Arg80 interference (Baker *et al.* 2013).

Shifts in gene expression patterns can also occur via mutation to cis-regulatory sites in a gene's control region, allowing a set of genes to functionally "swap" between TRs to elicited new expression patterns. These cis rewiring events can require tens to hundreds of mutations across the genome and various mechanisms have been put forth to explain how this occurs (Britten and Davidson 1971; Gasch *et al.* 2004; Borneman *et al.* 2007; Bourque *et al.* 2008; Haase *et al.* 2021). One notable example is the parallel evolutionary gains of Mcm1 cis-regulatory sites in ribosomal protein genes in fungi (Tuch, Galgoczy, *et al.* 2008; Sorrells *et al.* 2018). In this case, frequent gains of Mcm1 cis sites were proposed to be facilitated by intrinsic cooperative activation between the TRs Rap1 and Mcm1, a property inherent due to their deeply conserved interactions with the general transcription factor TFIID (Sorrells *et al.* 2018). In this scheme, suboptimal Mcm1 may have been selected for and subsequently optimized nearby ancestrally present Rap1 binding sites. Intriguingly, we observed Rap1 binding sites in a subset of *Hanseniaspora* FEL histone H2A–H2B control regions. Perhaps these may represent lingering transitional states between the Spt10-mode and Mcm1-mode, where intrinsic cooperative activation aided in replacement of Spt10. However, we did not find evidence of Rap1 binding sites in the histone cluster control regions of *Hanseniaspora* SEL species, or any other Saccharomycotina species, thus it is not clear if a mechanism of intrinsic cooperative activation allowed for the exchange between an ancestral Spt10-mode to the derived Mcm1-mode. Nonetheless, more recent work demonstrated that cooperativity between TRs can emerge with ease over evolutionary time (Fowler *et al.* 2023).

In other cases, the same set of genes are regulated by different TRs in different lineages, but illicit subtle differences in gene expression patterns. For example, the transcriptional induction of *GAL1*, *GAL7*, and *GAL10* by galactose is mediated via the TR Gal4 in *S. cerevisiae*, whereas in *Candida albicans*, transcriptional induction of this set of genes is instead thought to be mediated by the TRs Rtg1/3 (Martchenko *et al.* 2007; Dalal *et al.* 2016). Analogous to what we observed, while the overarching dynamics are achieved by different TRs (transcriptional induction by galactose or, in our case, cell cycle expression of histones), important differences exist between the 2 swapped TR-modes. In the example of Rtg1/3-to-Gal4 swap, Gal4 regulated species exhibit a characteristic all-or-nothing expression, whereas Rtg1/3 regulated species show a more graded expression response to galactose (Biggar 2001; Dalal *et al.* 2016). Interestingly, deletion of the galactose permease, Gal2 or the negative co-regulator Gal80, switches *S. cerevisiae*'s all-or-nothing response to graded transcriptional induction

(Biggar 2001; Hawkins and Smolke 2006), suggesting that the more subtile differences in gene expression patterns emerge from species-specific proteins interaction networks. Indeed, the regulatory decoupling of histone and DNA synthesis in *H. uvarum* is strikingly similar to the behavior of *hir* mutants in *S. cerevisiae*, in which hydroxyurea-mediated repression of histone gene expression likely depends on direct recruitment of the HIR complex to the histone control region via the N-terminal region of Hpc2 (Vishnoi *et al.* 2011). Remarkably, the gene *HPC2* was lost ancestrally in the *Hanseniaspora* FEL (Steenwyk *et al.* 2019). Moreover, in *S. cerevisiae* Spt10 recruits the HIR complex to histone promoters outside of S-phase, thereby repressing histone gene expression (Kurat, Lambert, *et al.* 2014). Thus, alongside the loss of Hpc2 and the rewiring of histone control regions to a Mcm1-mode, it would seem that two critical histone gene repression mechanisms were lost in the ancestor of the *Hanseniaspora* FEL. It is an appealing idea that loss of histone gene repression mechanisms may be have been adaptive to a lifestyle of rapid growth—ensuring histones are made without any delay—however, we cannot exclude the possibility that these changes are a result of relaxed selection on the timing of histone gene expression and consequently neutral evolution of Mcm1 cis-regulatory sites and loss of histone gene repression mechanisms. Future work will be needed to uncover the temporal dynamics of DNA replication and other key cell cycle events at the single-cell level in *Hanseniaspora* FEL to determine if the changes in histone gene expression patterns were a result of adaptive evolution or whether this regulatory scheme evolved neutrally.

Given the uniqueness of genes absent from the *Hanseniaspora* FEL (Fig. 6h), we believe that it can be a future model for cell biology for understanding how typically essential processes function in the absence of key cell cycle regulators. Moreover, framing future studies within an evolutionary cell biology framework (Helsen *et al.* 2023) will aid in understanding the biochemical and molecular details of the evolution of core histone gene regulation in *Hanseniaspora* and budding yeast more broadly.

## Data availability

All yeast strains and plasmids are available upon request to Jef D. Boeke (Jef.Boeke@nyulangone.org). Raw RNA sequencing data were deposited to Sequence Read Archive (SRA) and are available under the Bioproject ID PRJNA987614 with accessions SRR25022105–SRR25022112.

Supplemental material available at GENETICS online.

## Acknowledgments

We thank Luciana Lazar-Stefanita for her helpful comments and support; Jürgen Heinisch for kindly providing strains, plasmids, and the transformation protocol for *Hanseniaspora uvarum*. We thank Gregory Goldberg for helpful comments on an early draft. We are grateful to Chris Hittinger and Antonis Rokas and the members of their laboratories for fruitful discussions of this work.

## Funding

An NSF Rules of Life grant supported this work: Epigenetics 2 (award number: MCB1921641) to JDB. JLS is a Howard Hughes Medical Institute Awardee of the Life Sciences Research Foundation.

## Conflicts of interest

JDB is a Founder and Director of CDI Labs, Inc., a Founder of and consultant to Neochromosome, Inc., a Founder, SAB member of,

and consultant to ReOpen Diagnostics, LLC, and serves or served on the Scientific Advisory Board of Logomix, Inc., Modern Meadow, Inc., Rome Therapeutics, Inc., Sample6, Inc., Sangamo, Inc., Tessera Therapeutics, Inc., and the Wyss Institute. JLS is a scientific advisor for WittGen Biotechnologies. JLS is an advisor for ForensisGroup, Inc.

## Author contributions

MABH conceptualized the project, performed the formal investigation, wrote the manuscript, and prepared figures. JLS provided comments and suggestions on the manuscript. JDB supervised the research and provided funding. All authors edited the manuscript.

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

*Editor: L. Cowen*