## [Peer Review File · Genetics]

Gene loss and cis-regulatory novelty shaped core histone gene evolution in the apiculate yeast *Hanseniaspora uvarum*

Max Haase, Jacob Steenwyk, and Jef Boeke

NOTE: The reviews and decision letters are unedited and appear as submitted by the reviewers.

In extremely rare instances and as determined by a Senior Editor or the EIC, portions of a review may be redacted. If a review is signed, the reviewer has agreed to no longer remain anonymous.

The review history appears in chronological order.

Review Timeline:

Submission Date:	2023-08-28
Editorial Decision:	2023-10-09
Revision Received:	2024-01-17
Accepted:	2024-01-18

October 9, 2023

RE: GENETICS-2023-306264

Dear Dr. Haase:

I am pleased to accept your manuscript entitled "**Gene loss and cis-regulatory novelty shaped core histone gene evolution in the apiculate yeast *Hanseniaspora uvarum***" for publication in GENETICS, pending minor revision.

Please submit your revision along with a response to the reviewers' concerns and suggestions, which can be viewed at the bottom of this email. Most important are to revise the manuscript to consider neutral evolution as the default hypothesis, ensure that the results are better positioned in context with the literature, and to revise the writing to enhance clarity and accessibility. I expect this can be done within 30 days.

Upon resubmission, please include:

1. A clean version of your manuscript;
2. A marked version of your manuscript in which you highlight significant revisions carried out in response to the major points raised by the editor/reviewers (track changes is acceptable if preferred);
3. A detailed response to the editor's/reviewers' comments and to the concerns listed above. Please reference line numbers in this response to aid the editors.

Additionally, please ensure that your revision is formatted for GENETICS: <https://academic.oup.com/genetics/pages/general-instructions>.

Follow this link to submit the revised manuscript: Link Not Available

Thank you for submitting your research to Genetics.

Sincerely,

Leah Cowen
Associate Editor
GENETICS

Approved by:
Meera Sundaram
Senior Editor
GENETICS

Reviewer comments:

Reviewer #1 (Comments for the Authors (Required)):

There is quite a lot to unravel in this in this interesting paper. The authors show that histone genes underwent gene duplication in an ancient yeast ancestor, and that one copy of the paralogs were lost in the Fast Evolving Lineage (FEL) of the *Hanseniaspora* lineage. They infer that the gene loss is associated with a switch from Spt10- to Mcm1-regulated expression of histone genes. Aberrant expression (i.e. Mcm1-driven) is toxic in *S. cerevisiae*. In addition, the authors characterize cell division in *Hanseniaspora uvarum*. They show that histone synthesis occurs independently of DNA synthesis, which is very different to *S. cerevisiae*. A mixture of experimental and bioinformatic approaches are used to greatly extend our understanding of histone regulation and cell division beyond the "model" yeast *S. cerevisiae*.

The manuscript is quite dense and is at times difficult to follow, possibly because a lot of material is presented. Some suggestions are provided below.

1. The term "Saccharomycotina" is used a few times but it is not well defined, and in general it is assumed that the reader has a very good prior understanding of yeast phylogeny. In addition, the results section dives straight into histone gene loss in

Hanseniaspora without first describing which species have duplicate copies. This could be at least partly addressed by moving some of Fig. 6A to the beginning of the manuscript, and describing the presence/absence of histone genes. Fig S1 should also be expanded to include more Saccharomycotina species. This would help with interpreting later figures, e.g. Fig. 2 and Fig. S5.

2. The discussion of the loss of introns appears suddenly in the conclusions, and should be described much earlier.
3. Saccharomyces eubayanus is in the figures but not the text. Presumably it is being used as a positive control, which should be explained.
4. Interpreting the text associated with Fig. 2 requires details from Fig S4. It is difficult to follow the assay used without Fig. S4, and the "weak complementation" described for Sch2A and Sch2B is crucial for the interpretations presented. It would therefore be useful to include Fig S4 in the main manuscript.
5. "Outgroup" is used to refer to Lipomyces and Tortispora (page 218) and then to "outgroup species" in Fig. 1 and Fig. S5. It is not always clear what species are included as "outgroups" in each analysis.
6. To my mind, the analysis in Lipomyces/Tortispora (from line 212) does not fit well with the rest of the paper, and confuses the main story.
7. Fig 2C: are rows 4 and 6 the same construct? It is not a problem to have repeats (likes rows 1 and 8) but they should be clear.
8. Fig. 2D: can you use some measurements of significance for similarities/differences in expression? The data is difficult to interpret because there is a lot of variation (especially for the final construct).
9. Fig. 3 legend refers to Fig. S8 and S9 which are not there (the S8 provided shows cell length). The raw RNA-seq data has been deposited to SRA, but some intermediate information (as well as Fig. 3C) would be useful.
10. This is personal preference, but I find that essentially combining results and discussion can complicate understanding, such as when the ancestral MADS gene is discussed around line 319. In addition. The discussion of HPC2 (line 408) is not well integrated with the rest of the manuscript.

Minor points

1. Check the grammar from lines 159-162: there appears to be incomplete sentences.
2. The term "HuvaPro" is not very clear. Are you referring to a specific promoter, or a specific strain construct?
3. Line 195: you don't "perform motif enrichments", but instead test for enrichment of motifs.
4. Line 413 refers to Fig. 5F; I think this should be Fig. 5H.

Reviewer #2 (Comments for the Authors (Required)):

Major Comments

Overall, this paper convincingly shows that the control region for the histone genes has undergone a change in regulation in the Hanseniaspora lineage. There are a few statements in the paper that the authors should revise to consider neutral evolution rather than adaptation as the default hypothesis.

Line 24. Rather than stating that the change they document is likely adaptive, the authors should consider an alternative hypothesis: that selection for the timing of histone gene expression was simply relaxed, permitting the alternative regulatory scheme to evolve neutrally. In any case, there is no evidence that the change was adaptive, so stating that adaption is "likely" is simply not justified. The incompatibility experiments of Figure 2 do not address whether the change was adaptive, as many incompatibilities among species can arise after an evolutionary change has occurred.

Line 464. Along the same lines as the above point, there is no evidence that there was "strong diversifying selection" driving the changes the authors document. Again, neutral evolution cannot be ruled out and, in the absence of evidence to the contrary, should probably be considered the default hypothesis.

Finally, "swaps" among regulators--resulting in differences in gene expression patterns--have been documented in many other cases in the fungal lineage, and it would strengthen the paper if the authors attempted to place their discoveries in this broader context.

Minor comments

L70-71 In general, genes encoded in budding yeast species lack introns (In *S. cerevisiae*, less than 5% of genes), and histones are not unusual in this respect.

L177 "divergence levels observed in the Hanseniaspora histones"

How are "levels of divergence" quantified? It appears the species tree is used to arrange the histone sequences in supplementary figures 2-3, and that appearance of overall divergence is qualitative. Consider rephrasing.

L193 Here and throughout the text consider changing "gene promoters" to "gene control regions." Many scientists use promoter to indicate only the DNA sequences that are directly bound by the general transcription factors and RNA polymerase.

L662: replace parenthetical word "citation" with the actual citation for the yeast gene order browser.

L270: include issue number and page numbers.

L564 include citation for yeast gene order browser.

No reference to figure S5B in the text.

Figure 2: UAS is an undefined acronym in the figure legend and the text. Please define, especially for non-yeast scientists.

Figure 4: E and F are out of order.

Figure 5B: figure axes are not defined.

Associate Editor comments:

Reviewer comments:

Reviewer #1 (Comments for the Authors (Required)):

There is quite a lot to unravel in this in this interesting paper. The authors show that histone genes underwent gene duplication in an ancient yeast ancestor, and that one copy of the paralogs were lost in the Fast Evolving Lineage (FEL) of the *Hanseniaspora* lineage. They infer that the gene loss is associated with a switch from Spt10- to Mcm1-regulated expression of histone genes. Aberrant expression (i.e. Mcm1-driven) is toxic in *S. cerevisiae*. In addition, the authors characterize cell division in *Hanseniaspora uvarum*. They show that histone synthesis occurs independently of DNA synthesis, which is very different to *S. cerevisiae*. A mixture of experimental and bioinformatic approaches are used to greatly extend our understanding of histone regulation and cell division beyond the "model" yeast *S. cerevisiae*.

We thank the reviewer for time to provide us with a thoughtful critique of our work. We have attempted to address the below concerns and suggestions and believe that we have produced a much-improved manuscript.

The manuscript is quite dense and is at times difficult to follow, possibly because a lot of material is presented. Some suggestions are provided below.

1. The term "Saccharomycotina" is used a few times but it is not well defined, and in general it is assumed that the reader has a very good prior understanding of yeast phylogeny. In addition, the results section dives straight into histone gene loss in *Hanseniaspora* without first describing which species have duplicate copies. This could be at least partly addressed by moving some of Fig. 6A to the beginning of the manuscript, and describing the presence/absence of histone genes. Fig S1 should also be expanded to include more Saccharomycotina species. This would help with interpreting later figures, e.g. Fig. 2 and Fig. S5.
2. The discussion of the loss of introns appears suddenly in the conclusions, and should be described much earlier.

We combine our response to 1&2 as our edits attempt to address them together.

We now begin the results section with our description of histone gene evolution in Saccharomycotina, reporting histone gene cluster copy number and the presence or absence of introns. This is shown in revised figure 1, which we present the "current" state of knowledge of the phyletic distribution of Spt10 histone gene regulation. Revised Figure 1 also now incorporates a simplified schematic demonstrating that histones are synthesized during S-Phase of the cell cycle – which we feel will prepare the reader better for experiments in Revised Figures 5 and 6.

See lines 126 – 142.

3. *Saccharomyces eubayanus* is in the figures but not the text. Presumably it is being used as a positive control, which should be explained.

The referee is correct, *S. eubayanus* histone genes and promoters are being used as a positive control.

We now make this clear on lines 221–223.

4. Interpreting the text associated with Fig. 2 requires details from Fig S4. It is difficult to follow the assay used without Fig. S4, and the "weak complementation" described for Sch2A and Sch2B is crucial for the interpretations presented. It would therefore be useful to include Fig S4 in the main manuscript.

We have now moved Fig S4 to the main manuscript, incorporating the data into Fig 1. We also provide the quantifications of 5-FOA resistance frequency as a more streamlined graph, aggregating the FEL and SEL histone together – demonstrating FEL histones H3 and H2A perform worse than the SEL counterparts.

See revised Figure 2 and associated text on line 205–243.

5. "Outgroup" is used to refer to *Lipomyces* and *Tortispora* (page 218) and then to "outgroup species" in Fig. 1 and Fig. S5. It is not always clear what species are included as "outgroups" in each analysis.

We now explicitly define outgroups for each analysis.

See lines 681–691 for MEME analysis that identified Mcm1 and Spt10-like motifs.

See lines 659–661 and 758–759 for histone cluster presence/absence as shown in Fig S1.

6. To my mind, the analysis in *Lipomyces*/*Tortispora* (from line 212) does not fit well with the rest of the paper, and confuses the main story.

We have opted to remove this analysis and associated figure based on the reviewer's comment.

7. Fig 2C: are rows 4 and 6 the same construct? It is not a problem to have repeats (likes rows 1 and 8) but they should be clear.

In this figure rows 4 and 6 (counting down) are distinct constructs. Row 4 is *H. uvarum* histones and histone control regions, with Mcm1 sites replaced by Spt10 sites. Whereas, row 6 is *H. uvarum* histones with *S. cerevisiae* histone control regions. Also rows 1 and 8 are distinct. Here row 1 is *S. cerevisiae* histones and *H. uvarum* histone control regions, with Mcm1 sites replaced by Spt10 sites. And row 8 is *S. cerevisiae* histone and histone control regions. We have modified the figure and figure legend to hopefully make these clearer.

See revised Figure 3D and lines 283–292.

8. Fig. 2D: can you use some measurements of significance for similarities/differences in expression? The data is difficult to interpret because there is a lot of variation (especially for the final construct).

We now provide a statistical analysis comparing the means of each construct (one-way ANOVA with multiple test correction). These analyses confirm our conclusions that deletion of the Mcm1 UAS lowers expression – significantly ($p > 0.05$) reduced mean GFP intensity – compared to other constructs (excluding the EV control). Also, we find that there are not statistically significant differences in mean GFP intensities between the Native Scer (HTA1) histone control region, the native *H. uvarum* histone control regions, and the *H. uvarum* histone control region with Spt10 UAS replacement. We now show each point (measured cell) in the data set.

See revised Figure 3E and lines 292–297 and 318–329.

9. Fig. 3 legend refers to Fig. S8 and S9 which are not there (the S8 provided shows cell length). The raw RNA-seq data has been deposited to SRA, but some intermediate information (as well as Fig. 3C) would be useful.

We corrected legend of Fig 3 (revised Figure 4).

We have modified our presentation of the RNA-seq data in the hopes of providing “intermediate information”. First, we have moved the volcano plot to the supplement (Revised Figure S5F) and have replaced it with a X-Y plot of transcript abundance in Revised Figure 4C. The main reasoning is that we think it shows better that the two strains have very few transcriptional changes – across the dynamic range of abundances. We also provide gene labels for genes which were significantly dysregulated – thus the same information is present as the volcano plot. We additionally provide an analysis of the correlation between each RNA-seq replicate (See revised Figure S5E) – again supporting our claim their transcriptomes are near identical. Lastly, we provide an excel sheet (Table S4) that lists each gene, its' transcript abundance from each replicate, and the statistics used in determining the differentially expressed genes.

See Revised Figure 4 and revised Figure S5. Lines 361–377. See Supplemental Table 4.

10. This is personal preference, but I find that essentially combining results and discussion can complicate understanding, such as when the ancestral MADS gene is discussed around line 319. In addition. The discussion of HPC2 (line 408) is not well integrated with the rest of the manuscript.

We have taken the decision to move some of these in-results-discussions to the *Discussion* section proper. In addition, based on reviewer's #2 comments we have overhauled the *Discussion* section; thus we refer the reviewer to the new section on lines 483–606.

Specifically, the ancestral MADS gene is discussed on lines 518–546.

Minor points

1. Check the grammar from lines 159-162: there appears to be incomplete sentences.

We reworded these sentences – see lines 207–211.

2. The term "HuvaPro" is not very clear. Are you referring to a specific promoter, or a specific strain construct?

Based on reviewer's #2 comments we have changed the name of HuvaPro to HuvaHCR (histone control region). See lines 248–249 and lines 303–304.

3. Line 195: you don't "perform motif enrichments", but instead test for enrichment of motifs.

We have corrected this error. See lines: 251–253

4. Line 413 refers to Fig. 5F; I think this should be Fig. 5H.

Due to other changes in the text, this was corrected.

Reviewer #2 (Comments for the Authors (Required)):

Major Comments

Overall, this paper convincingly shows that the control region for the histone genes has undergone a change in regulation in the *Hanseniaspora* lineage. There are a few statements in the paper that the authors should revise to consider neutral evolution rather than

adaptation as the default hypothesis.

We thank the reviewer for their time taken to critique our work and for their helpful suggestions and comments. We are especially thankful for the suggestion to frame our discussion in the context of previous work on trans regulator swaps.

Line 24. Rather than stating that the change they document is likely adaptive, the authors should consider an alternative hypothesis: that selection for the timing of histone gene expression was simply relaxed, permitting the alternative regulatory scheme to evolve neutrally. In any case, there is no evidence that the change was adaptive, so stating that adaptation is "likely" is simply not justified. The incompatibility experiments of Figure 2 do not address whether the change was adaptive, as many incompatibilities among species can arise after an evolutionary change has occurred.

Line 464. Along the same lines as the above point, there is no evidence that there was "strong diversifying selection" driving the changes the authors document. Again, neutral evolution cannot be ruled out and, in the absence of evidence to the contrary, should probably be considered the default hypothesis.

As both comments are similar, we provide a single response below.

We removed this verbiage from our abstract as we have not shown the changes were adaptive. As a result, we rewrote much of the abstract as can be seen on lines 17–33.

Further, we have removed the language from our manuscript that strongly advocated for adaptive explanations. We have instead provided a section in the discussion/conclusions to make clear that we cannot prove adaptive explanations given our data and that a neutral explanation is just as likely.

See lines: 592–599.

Finally, "swaps" among regulators--resulting in differences in gene expression patterns--have been documented in many other cases in the fungal lineage, and it would strengthen the paper if the authors attempted to place their discoveries in this broader context.

We have rewritten our discussion/conclusion section to discuss our results in the context suggested by the reviewer.

Please see the revised discussion on lines 483–606 and specifically the parts on regulatory "swaps" on lines 522–599.

We thank the reviewer for encouraging us to place our results in this context.

Minor comments

L70-71 In general, genes encoded in budding yeast species lack introns (In *S. cerevisiae*, less than 5% of genes), and histones are not unusual in this respect.

We have now made clear that it is not unusual that histone genes lost their introns and is a common trend for genes in budding yeast.

See lines 141–142.

L177 "divergence levels observed in the *Hanseniaspora* histones"

How are "levels of divergence" quantified? It appears the species tree is used to arrange the histone sequences in supplementary figures 2-3, and that appearance of overall divergence is qualitative. Consider rephrasing.

We now provide level of divergence of *Hanseniaspora* histones as Amino Acid substitutions per site as determined from ML phylogenetic inferences of histone protein sequences. We provide in the supplement the alignment and ML phylogenetic tree of histone H2A – for simplicity and removing redundancy, we do not show the alignments or ML trees for H2B, H3, and H4. Instead, we summarize our results in two graphs that plot the Amino Acid substitutions per site of *Hanseniaspora* histones compared to all other species examined.

Please see revised Figure S2 and lines 205–207

L193 Here and throughout the text consider changing "gene promoters" to "gene control regions." Many scientists use promoter to indicate only the DNA sequences that are directly bound by the general transcription factors and RNA polymerase.

We have replaced our usage of "promoter" to "gene control region" as suggested. Throughout the text we now refer to "histone control regions".

We define this on lines 248–249.

L662: replace parenthetical word "citation" with the actual citation for the yeast gene order browser.

Fixed.

L270: include issue number and page numbers.

We are not sure which reference this refers to as no citation is provided on the original line 270. We scanned over of references and have ensured the appropriate information is provided.

L564 include citation for yeast gene order browser.

Citation has been added.

No reference to figure S5B in the text.

No longer relevant due to other changes.

Figure 2: UAS is an undefined acronym in the figure legend and the text. Please define, especially for non-yeast scientists.

We had originally defined UAS in the introduction (see line 73) we now modified this to put the first letter in uppercase to more clearly indicate the acronym. We also define the acronym in the results and figure legends. See lines 275 and 304.

Figure 4: E and F are out of order.

We have swapped the order of E and F in the figure.

See revised Figure 5.

Figure 5B: figure axes are not defined.

We apologize for not clearly defining the axes, as we originally opted to simplify the presentation of the flow cytometry data. We now provide the actual x-axis – either Sytox Green intensities or mNeonGreen intensities – and the y-axis, normalized counts of cells (we leave this unlabeled as the total area under each curve = 1).

See revised Figure 6.

January 18, 2024

RE: GENETICS-2023-306264R1

Dr. Max A. B. Haase
New York University
Institute for Systems Genetics
435 East 30th Street
Room 923B
New York, New York 10016

Dear Dr. Haase:

Congratulations! We are delighted to inform you that your manuscript entitled "**Gene loss and cis-regulatory novelty shaped core histone gene evolution in the apiculate yeast *Hanseniaspora uvarum***" is acceptable for publication in GENETICS. Many thanks for submitting your research to the journal.

To Proceed to Production:

Add oupsupport@scipris.com and genetics.oup@novatechset.com (or the domains @scipris.com and @novatechset.com) to your email program's "safe senders" list. You will be contacted by both at various points during the production process.

1. Format your article according to GENETICS style, as discussed at <https://academic.oup.com/genetics/pages/general-instructions>. Note that the Methods section should be placed before the Results section. Ensure that you comply with data and community resource citation guidelines (<https://academic.oup.com/genetics/pages/general-instructions#Data-Policy>).
2. Upload your final files at <https://genetics.msubmit.net>.
3. Your currently-accepted manuscript (unedited, as submitted, reviewed, and accepted) will be published at GENETICS and deposited into PubMed as an Advance Access article. Notify sourcefiles@thegsajournals.org before signing your license if you do not wish to publish your article via Advance Access.
4. We invite you to submit an original color figure related to your paper for consideration as cover art. Please email your submission to the editorial office or upload it with your final files. You can submit a small-sized image for evaluation, and if selected, the final image must be a TIFF file 2513px wide by 3263px high (8.375 by 10.875 inches; resolution of 600ppi). Please avoid graphs and small type.
5. After files are sent to Oxford University Press we use SciPris to manage article licensing and payment. If you do not have a SciPris account, you will receive an email from no-reply@scipris.com to sign up to use Oxford University Press' author portal. After logging in, follow the online instructions to sign your licence and arrange any payment due.

If you have any questions or encounter any problems while uploading your accepted manuscript files, please email the editorial office at sourcefiles@thegsajournals.org.

Sincerely,

Leah Cowen
Associate Editor
GENETICS

Approved by:
Meera Sundaram
Senior Editor
GENETICS

Review comments (if applicable):